# A New Surgical Site Infection Risk Score: Infection Risk Index in Cardiac Surgery

**DOI:** 10.3390/jcm8040480

**Published:** 2019-04-09

**Authors:** Juan Bustamante-Munguira, Francisco Herrera-Gómez, Miguel Ruiz-Álvarez, Ana Hernández-Aceituno, Angels Figuerola-Tejerina

**Affiliations:** 1Cardiac Surgery, Hospital Clínico Universitario de Valladolid, 47003 Valladolid, Spain; jbustamantemunguira@gmail.com; 2Anatomy and Radiology, Pharmacology and Therapeutics, Faculty of Medicine, University of Valladolid, 47003 Valladolid, Spain; 3Nephrology, Hospital Virgen de la Concha, 49022 Zamora, Spain; 4Prevention and Control of Infection, Hospital Universitario de La Princesa, 28006 Madrid, Spain; rualmiguel@me.com (M.R.-Á.); anahdez989@gmail.com (A.H.-A.); angels.figuerola@gmail.com (A.F.-T.)

**Keywords:** surgical site infection, patient outcome, cardiac surgical procedures, scoring systems

## Abstract

Various scoring systems attempt to predict the risk of surgical site infection (SSI) after cardiac surgery, but their discrimination is limited. Our aim was to analyze all SSI risk factors in both coronary artery bypass graft (CABG) and valve replacement patients in order to create a new SSI risk score for such individuals. A priori prospective collected data on patients that underwent cardiac surgery (*n* = 2020) were analyzed following recommendations from the Reporting of studies Conducted using Observational Routinely collected health Data (RECORD) group. Study participants were divided into two periods: the training sample for defining the new tool (2010–2014, *n* = 1298), and the test sample for its validation (2015–2017, *n* = 722). In logistic regression, two preoperative variables were significantly associated with SSI (odds ratio (OR) and 95% confidence interval (CI)): diabetes, 3.3/2–5.7; and obesity, 4.5/2.2–9.3. The new score was constructed using a summation system for punctuation using integer numbers, that is, by assigning one point to the presence of either diabetes or obesity. The tool performed better in terms of assessing SSI risk in the test sample (area under the Receiver-Operating Characteristic curve (aROC) and 95% CI, 0.67/055–0.76) compared to the National Nosocomial Infections Surveillance (NNIS) risk index (0.61/0.50–0.71) and the Australian Clinical Risk Index (ACRI) (0.61/0.50–0.72). A new two-variable score to preoperative SSI risk stratification of cardiac surgery patients, named Infection Risk Index in Cardiac surgery (IRIC), which outperforms other classical scores, is now available to surgeons. Personalization of treatment for cardiac surgery patients is needed.

## 1. Introduction

### 1.1. Rationale

Patients who undergo operations can present numerous complications, among which infections stand out owing to their frequency and severity [1,2,3]. Surgical site infection (SSI) is the second most common location of healthcare-related infections; the incidence of SSI after cardiac surgery ranges from 1.1–7.9% and is associated with high morbidity and mortality as well as significant increase in healthcare costs [4,5,6,7]. Importantly, although real figures of incidence vary depending on the type of surgical procedure, study design, and definitions used to classify the infection, the risk of this complication is a concern among physicians [8,9,10,11,12]. Improvements in operating procedures over the last decade should also be considered, as well as the profile of patients who undergo cardiac surgery: patients are older, have more comorbidities, and have more complex cardiac pathologies. Furthermore, special attention should be paid to the situation of diabetics and obese patients: (1) diabetics with preoperative hyperglycemia have an increased risk of SSI after cardiac surgery, with a significant influence of microcirculatory abnormalities [10], and (2) in obese patients, decreased blood flow in the adipose tissue is also associated with higher rates of deep SSI after cardiac surgery [7].

The European System for Cardiac Operative Risk Evaluation (EuroSCORE) is the most frequently used scale for patient risk stratification in Europe. This scale is used to predict mortality and the appearance of complications, particularly kidney failure, and its performance in coronary intervention settings has been well-studied [13]. Nevertheless, more specific soring systems are used to stratify SSI risk. The Centers for Disease Control and Prevention (CDC) has proposed a method called the National Nosocomial Infections Surveillance (NNIS) risk index in an attempt to reduce SSI risk in surgical procedures [14]. This index combines the following three variables: level of contamination of the surgical wound, the American Society of Anesthesiologists (ASA) pre-anesthesia score, and the duration of the surgical procedure [15]. One point is assigned if the patient has an ASA score of III, IV, or V; one point if the wound is contaminated or dirty; and another point if the surgical procedure lasts longer than 75% (P75) of the total surgical duration. Consequently, the NNIS risk index score ranges from 0 (the lowest SSI risk) to 3 (the highest risk). However, although the NNIS risk index is internationally accepted, its predictive power has certain limitations in cardiac surgery setting, given the prolonged duration of common surgical procedures [16].

The Australian Clinical Risk Index (ACRI) of patients destined for coronary artery bypass graft (CABG) combines diabetes mellitus (DM) and body mass index (BMI), correlating these factors with a gradual increase in SSI risk. It assigns 1 point to DM diagnosis, 1 point if BMI is 30–34.9, and 2 points if BMI is 35 kg/m^2^ or more. ACRI offers an improved risk estimation compared to the NNIS risk index [17], and it has been validated in both the United States [18] and recently by our European population group [16]. Although its discrimination falls significantly for patients undergoing valve replacement, other factors should be taken into account [16,18].

### 1.2. Aim

Various scoring systems have been designed with the aim of predicting the SSI risk after cardiac surgery, but their discriminatory abilities are limited, and their use is still unfamiliar for all physicians. Many of risk factors for developing SSI such as kidney failure, vascular disease, surgical procedure type, re-operation for bleeding, need for transfusion, etc. have been omitted from available scores. This work presents an analysis of preoperative, perioperative, and postoperative SSI risk factors in both CABG and valve replacement patients. Based on this analysis, a new risk score is proposed and its suitability is assessed in order to improve physicians’ decision-making processes, with the primary aim being the personalization of patient treatment.

## 2. Materials and Methods

### 2.1. Real-World Study Characteristics

This study was based on automated electronic medical record (EMR) data feeds that were conceived, performed, and reported according to the Reporting of studies Conducted using Observational Routinely collected health Data (RECORD) guidelines [19]. All patients that underwent major cardiac surgery were included consecutively with the intention of developing a new SSI risk index (a priori prospective collected data) [20]. Individuals under the age of 18 years, patients undergoing minithoracotomy, and/or those operated on without extracorporeal circulation were excluded. An integrated delivery network of physician and nursing records, operative reports, summary of labs, and microbiological analyses provided daily monitoring information of patients from admission to discharge, and from readmissions associated with a complication and/or infection during the year following surgery. This information was sourced from a European level 3 university teaching hospital via the Spanish system of healthcare-associated infection surveillance (programa Indicadores Clinicos de Mejora Continua de la Calidad (INCLIMECC)) [21,22], and was retrieved in the final anonymized dataset.

The date of sign or symptom onset, type of SSI, and microbiological results in exudate cultures (bioMérieux Vitek® 2 system, Marcy-l’Étoile, Lyon, France) were available. Admissions were classified into two groups: (1) preoperative stays less and (2) more than 48 h (minimum hospital stay for hospital-acquired infections). Antibiotic prophylaxis with intravenous cefazolin (1000 mg every 4 h during the operation), or vancomycin (1000 mg before surgery and 500 mg afterwards for 48 h) in patients allergic to beta-lactams were also recorded. Patients that died during their hospital stay were not excluded from the analysis.

In this study, a harmonized and predefined variable-guided data collection procedure was guaranteed. Variables accounted for a total of 48 preoperative, perioperative, and postoperative SSI risk factors common for both CABG and valve replacement patients, including two BMI categories (less than or equal to, and greater than 30 kg/m^2^) and two duration of surgery categories (less than or equal to, and greater than P75). The CDC definition of SSI to standardize data collection for the National Healthcare Safety Network (NHSN) (also called the NNIS program) leads to the classification of SSI as either incisional, which can be superficial (skin or subcutaneous tissues) or deep (facial and muscle layers), or organ/space SSI (mediastinitis) [23]. In addition, deep sternal wound infection (DSWI) (osteomyelitis and mediastinitis) was included when one of the following criteria was presented [24]: (1) an organism was isolated from cultures of mediastinal tissue or fluid obtained invasively; (2) evidence of mediastinitis was seen during operation; or (3) at least one of the following signs or symptoms—fever (>38 °C), chest pain, or sternal instability—accompanied purulent discharge from the mediastinum or organisms isolated from blood or discharge from mediastinal area or mediastinal widening on an imaging test.

Data collection was similarly performed for all study participants that were included in two periods: (1) from 1 January 2010 to 31 December 2014, to identify variables predicting SSI (training sample); and (2) from 1 January 2015 to 31 December 2017, to validate the results obtained from the training sample (test sample).

Our Hospital Ethics Committee approved the study and waived the need for informed consent from the patients. However, the patients gave their written consent to store their data in an anonymous way in the final dataset destined for scientific treatment, in accordance with Spanish legal regulation of personal privacy matters.

### 2.2. Statistical Analyses

Data were expressed as number or percentage (rate) of patients, mean ± standard deviation (SD), or median with interquartile range. Categorical variables such as patients’ preoperative clinical characteristics, operative data, and postoperative complications were compared using χ2 test or Fisher’s exact test when appropriate, and continuous variables using Student’s *t*-test or Mann–Whitney U-test. Statistical significance was set at *p* < 0.05.

A logistic regression model was constructed with all variables showing statistically significant association in the univariate analysis, as well as with variables considered relevant for the development of the new SSI risk index, even where no association was found. Binary independent variables were presented in odds ratio (OR) with their corresponding 95% confidence interval (95% CI). A scaled exclusion strategy was used until the definitive model was reached. The model with the greatest discriminative (predictive) capacity, well calibrated (Hosmer–Lemeshow test), stable, and the most parsimonious (i.e., by including the smaller number of parameters) was selected. The new SSI risk score, herein referred to as Infection Risk Index in Cardiac surgery (IRIC), was then constructed with independent risk factors identified from the chosen multivariable logistic regression model. IRIC was conceived using a summation system for punctuation using integer numbers for each variable (yes = 1 point, no = 0 points).

Independent discrimination for IRIC, NNIS risk index, ACRI, and EuroSCORE-1 was assessed by calculating the area under the Receiver-Operating Characteristic curve (aROC) corresponding to each of these scoring systems on the training sample and the test sample. For each scoring system, the statistical significance of the difference between the areas under the ROC curves derived from the training sample and the test sample was then evaluated using the method of Hanley and McNeil (1982) [25]. Finally, the statistical significance of the difference between the areas under the ROC curves corresponding to IRIC, NNIS risk index, ACRI, and EuroSCORE-1 derived from the test sample (the same cases) was assessed using Hanley and McNeil’s method (1983) [26].

Statistical analyses were performed using the Statistical Package for the Social Sciences (SPSS version 24.0.; SPSS Inc, Chicago, IL, USA) and the fitting of regression with R programming language version 3.4.1. (R foundation for statistical computing, Vienna, Austria).

## 3. Results

### 3.1. Study Participants and Surgical Site Infection Occurrence

A total of 1298 procedures were included, with the following distribution (% and *n*): valve replacement, 73/944; CABG, 17/221; and mixed, 10/133. The demographic and clinical characteristics of the study participants are presented in Table 1.

Fifty-seven percent of study participants were male individuals with an average age of (mean ± SD) 70 ± 12 years. No differences were observed between the patients in the training and test samples (Appendix A), which included patients with DM (27%), chronic kidney disease (9%), peripheral artery disease (7%), and obesity (5%). Almost all surgical procedures were scheduled (95%). Clean surgeries represented 91%, and 88% of the patients had an ASA score of III or IV. Mean postoperative stay was 16 ± 15 days. Mortality was 6%, and no differences between patients developing and not developing SSI were observed.

SSI incidences (rate) were as follows (% and 95% CI): overall, 4.6/3.5–5.7; incisional, 2.8/1.9–3.7; mediastinitis, 1.5/0.8–2.2; valve replacement, 3.9/2.7–5.1; CABG, 7.2/3.8–10.6; and mixed, 5.3/1.5–9.1. Among all recorded SSI (*n* = 60), wound cultures were commonly positive (95%). Signs and symptoms compatible with SSI such as fever or chest pain and purulent wound drainage were frequent. Isolated microorganisms included gram-positive (61%) and gram-negative bacteria (37%) as well as fungi (2%), most frequently: *Staphylococcus aureus* (16%), *Staphylococcus epidermidis* (12%), *Pseudomonas aeruginosa* (6,6%), *Escherichia coli* (5%), and *Serratia marcescens* (5%) (Table 2). In 21 (36%) patients, more than one strain was identified.

Although EuroSCORE-1 punctuations (mean ± SD, 7.2 ± 7.9) showed significant statistical differences between patients with and without SSI (10.5 ± 12 vs. 7 ± 7.7; *p* < 0.05), SSI incidence according to this scale was, respectively, in low (less than 2), intermediate (between 2 and 5), and high (more than 5) punctuation groups (in %): 4.3, 3.4, and 5.7. According to the NNIS risk index, the incidence was (NNIS category, % valve replacement/% CABG/75P duration of surgery (300 min)): 0, 0.9/0.0/2.1; 1, 2.5/3.1/3.6; 2, 3.6/5.6/7.4; 3, 0.0/0.0/0.0. The incidence for ACRI was (ACRI category, %): 0, 2.7; 1, 4.3; 2, 9.8; 3, 14.3 (χ2 test (trend) 18.89; *p* < 0.05).

### 3.2. Calculation and Construction of IRIC

Univariate analysis showed a significant association of SSI with age, diabetes, obesity, peripheral artery disease, and duration of surgery. Nevertheless, among these variables, in multivariate analysis the following were associated independently and significantly with SSI (OR and 95% CI): diabetes, 3.30/2.18–5.67; obesity, 4.50/2.22–9.28 (Table 3). Non-significant association was found with duration of surgery (1.00/1.00–1.01). Renal failure and respiratory pathologies were not associated with SSI. The Hosmer–Lemeshow test confirmed the calibration of the logistic regression model (*p* = 0.523).

By assigning one point to the presence of either diabetes or obesity, IRIC-based SSI incidence was, respectively, for the conformed categories 0, 1, and 2 (in %): 2.5, 8.4, and 28.6 (χ2 test (trend) 40.44; *p* < 0.05).

Figure 1 and Table 4 present the independent discriminative capacities for IRIC, NNIS risk index, EuroSCORE-1, and ACRI in the training sample (aROCs and 95%CI): 0.70/0.63–0.78; 0.59/0.52–0.67; 0.58/0.50–0.67; 0.67/0.59–0.74. These scoring systems’ discrimination levels were similar in both the training and test samples [25]. Figure 2, Table 5 and Appendix A present the discrimination levels of IRIC (aROC and 95% CI, 0.66/055–0.76), NNIS risk index (0.61/0.50–0.71), ACRI (0.61/0.50–0.72), and EuroSCORE-1 (0.48/0.36–0.61) in the test sample and the differences between the areas under the ROC curves corresponding to these scoring systems [26].

## 4. Discussion

### 4.1. Key Findings

This real-world data study proposes a new simple and non-invasive SSI risk score for both CABG and valve replacement patients. The new score, named IRIC, was constructed after an analysis of preoperative, perioperative, and postoperative risk factors, and outperforms other well-known scales. This method substantially improves the monitoring of SSI, with priority given to personalizing the treatment of high-risk patients.

In recent years, several scoring systems have been developed to predict SSI risk in patients who undergo cardiac surgery [13,14,17]. However, these predictive models were developed from a series of patients who underwent various surgical procedures or from preselected CABG cohorts. Unfortunately, these models’ discrimination abilities are limited, primarily due to the complex pathogenesis of SSI, which involves specific comorbidities, periprocedural factors, and postoperative complications.

The NNIS risk index fails to present an effective discriminative capacity for patients undergoing cardiac surgery because these patients are highly similar to other operated patients [16]. Indeed, most cardiac surgery procedures are clean surgeries, and the ASA score is generally high, so the NNIS risk index is only capable of categorizing patients into two groups, those with more than 75P duration of surgery and those with 75P duration of surgery or less. In both cases, predictive properties of EuroSCORE and ACRI are equal to or slightly higher than that of the NNIS risk index. Importantly, IRIC and the other scores indicate that the risk of developing SSI increases when risk factors combine, with diabetic and obese patients’ incidences being 4 times that of patients presenting only one of these risk factors.

Although preoperative, perioperative, and postoperative risk factors were analyzed, IRIC may be considered as a preoperative tool; only the preoperative variables, diabetes and obesity, were significantly associated to SSI risk, compared to the factor of duration of surgery, which did not reach significant statistical association. This characteristic of IRIC should stress efforts for developing more user-friendly scoring systems that are more easily accessible and understandable to surgeons as compared to more complex tools. This study presents a two-variable score using an integer number-summation system for punctuation that outperforms the other available scoring systems owing to its analytic approach (although other approaches may be tried).

Non-significant association for duration of surgery was observed and this variable was not considered in the final tool. Again, complex relations between risk factors for and the occurrence of SSI prevent adequate treatment of high-risk patients submitted to cardiac surgery. Scoring systems based on variables belonging to different stages (e.g., preoperative and perioperative) are likely difficult to realize. However, clinicians must remain vigilant to all potential detrimental factors such as age [27], prolonged cross-clamp time [28], preoperative kidney dysfunction [29], low hematocrit [30], etc. Remarkably, diabetes mellitus and obesity are well-known factors of poor prognosis for patients who undergo cardiac surgery [31,32,33]. It is possible that prophylaxis with preemptive antibiotic treatment, an adequate metabolic control, and monitoring of such patients may contribute to combat SSI.

### 4.2. Strengths and Limitations

Real-world data scoring systems are crucial tools for the effective management of patients [34]. However, clinicians and researchers need to be aware of the positives and negatives of the proposed tools. As a starting point for more rigorous studies, analyses of propension originating from real-world observational sources are destined to improve decision-making processes [35], although questions about the quality of the evidence may arise [36]. Indeed, due to the observational nature of this study, its results should be interpreted with caution, allowing for differences among activities in other cardiac surgery centers and in different regions. In response to the current popularity of implementing research in real-world settings, a selection of the most appropriate study design must be accompanied by a clear definition of the intended application for the assessed data [37]. Our aim was to propose a new SSI scoring system, for which an a priori prospective collected data study was employed [20].

## 5. Conclusions

This study proposes a two-variable score for the preoperative stratification of cardiac surgery patients according to their SSI risk after surgery. The proposed tool, named IRIC, outperforms other routinely used scoring systems. The presented results may aid in personalizing the treatment of patients needing cardiac operation (e.g., prophylaxis with preemptive antibiotic treatment and closer monitoring of high-risk patients).

Regardless of IRIC’s possible applications, the more efficient management of patients requires the ability to use scores in clinical decision-making.

## Figures and Tables

**Figure 1 jcm-08-00480-f001:**
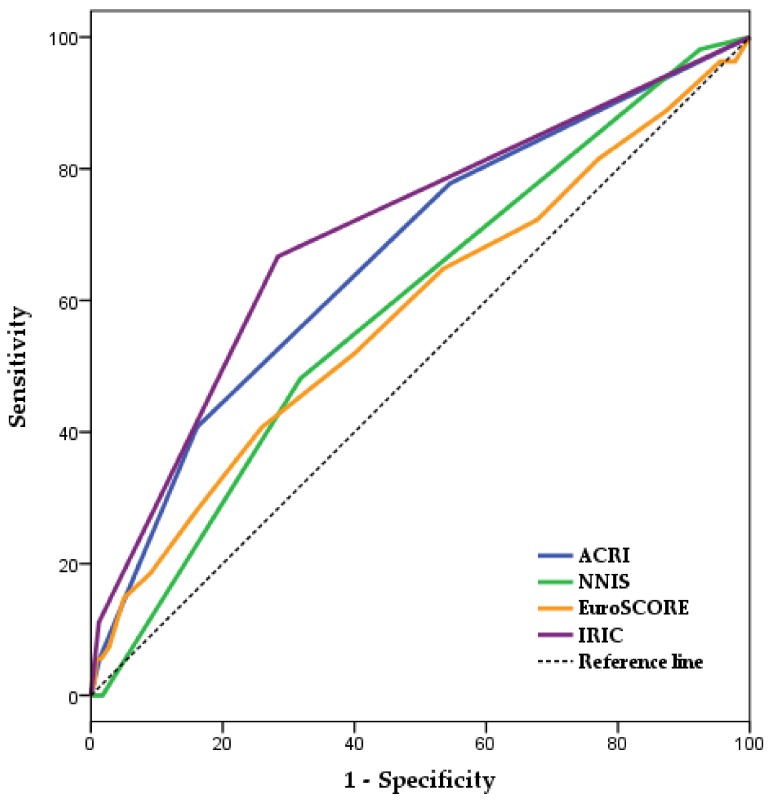
Independent aROC corresponding to IRIC, ACRI, NNIS risk index, and EuroSCORE-1 for the prediction of SSI in cardiac surgery patients (training sample). ACRI, Australian Clinical Risk Index; aROC, area under the Receiver-Operating Characteristic curve; CI, confidence interval; EuroSCORE, European System for Cardiac Operative Risk Evaluation; IRIC, Infection Risk Index in Cardiac surgery; NNIS, National Nosocomial Infections Surveillance.

**Figure 2 jcm-08-00480-f002:**
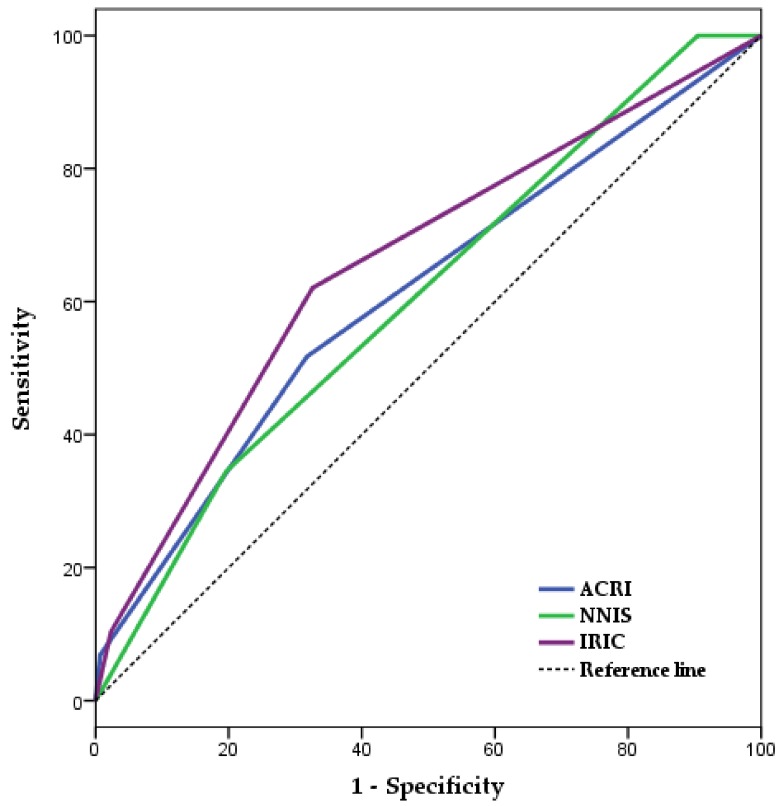
Comparisons of aROC corresponding to IRIC, NISS risk index and ACRI to evaluate differences between these scoring systems (test sample). ACRI, Australian Clinical Risk Index; aROC, area under the Receiver-Operating Characteristic curve; CI, confidence interval; IRIC, Infection Risk Index in Cardiac surgery; NNIS, National Nosocomial Infections Surveillance.

**Table 1 jcm-08-00480-t001:** Characteristics of study participants.

Variable	No SSI, *n* = 1238	SSI, *n* = 60	*p*
**Preoperative factors**			
Age in years (mean ± SD)	68 ± 12	72 ± 10	0.042
Males (*n*, %)	706 (57)	29 (48)	0.186
Diabetes (*n*, %)	314 (25)	32 (53)	<0.001
Obesity or BMI >30 kg/m^2^ (*n*, %)	54 (4)	11 (18)	<0.001
Peripheral artery disease (*n*, %)	85 (7)	9 (15)	0.025
Chronic kidney disease (*n*, %)	107 (9)	4 (7)	0.405
**Intraoperative factors**			
Emergency surgery (*n*, %)	84 (7)	4 (7)	0.786
Type of surgery (*n*, %):			0.099
CARD	907 (73)	37 (62)	
CABG	205 (17)	16 (27)	
CARD + CABG	126 (10)	7 (12)	
Duration of surgery in min (mean ± SD)	281 ± 81	303 ± 75	0.034
Aortic clamping time in min (mean ± SD)	67 ± 37	74 ± 39	0.191
Total CPB in min (mean ± SD)	96 ± 50	102 ± 58	0.426
Appropriate prophylaxis (*n*, %)	269 (22)	12 (22)	0.552
**Risk score**			
75P duration of surgery or 300 min (*n*, %)	402 (33)	29 (48)	0.009
Clean surgery (*n*, %)	1129 (91)	56 (93)	0.845
ASA score ≥ III (*n*, %)	1088 (88)	56 (93)	0.145
NNIS risk index (*n*, %):			0.011
0	93 (8)	2 (3)	
1	742 (60)	28 (47)	
2	377 (31)	30 (50)	
3	26 (2)	0 (0)	
**Postoperative factors**			
Reoperation for bleeding (*n*, %)	51 (4)	8 (13)	0.004
Length of hospital stay in days (mean ± SD):			
Before surgery	6 ± 6	5 ± 4	0.152
After surgery	16 ± 15	17 ± 14	0.501
Total in-hospital stay	21 ± 17	22 ± 15	0.925
Hospital mortality	72 (6)	6 (10)	0.133

ASA, American Society of Anesthesiologists; BMI, body mass index; CABG, coronary artery bypass graft; CARD, cardiac valve surgery; CPB, cardiopulmonary bypass; NNIS, National Nosocomial Infections Surveillance; P75, 75th percentile; SSI, surgical site infection.

**Table 2 jcm-08-00480-t002:** Isolated microorganisms from patients with SSI or bacteremia.

	SSI (*n*, %)	Bacteriemia (*n*, %)
*Candida albicans*	1 (1.6)	1 (7)
*Bacteroides caccae*	1 (1.6)	0
*Corynebacterium* spp.	0	1 (7)
*Enterococcus faecalis*	2 (3.3)	3 (21)
*Escherichia coli*	3 (5)	5 (36)
*Klebsiella pneumoniae*	2 (3.3)	0
*Proteus mirabilis*	1 (1.6)	1 (7)
*Pseudomonas aeruginosa*	4 (6.6)	1 (7)
*Serratia marcescens*	3 (5)	6 (43)
*Staphylococcus aureus*	8 (14)	6 (43)
*Staphylococcus aureus* methicillin-resistant	2 (3.3)	0
*Staphylococcus epidermidis*	7 (12)	1 (7)
*Staphylococcus coagulase*-negative	2 (3.3)	0
*Streptococcus* spp.	1 (1.6)	1 (7)
Polymicrobial	21 (34)	2 (14)
Without germens	3 (5)	0

SSI, surgical site infection.

**Table 3 jcm-08-00480-t003:** Independent SSI-associated factors for both CABG and valve replacement patients.

Variables	OR	95% CI	*p*
Diabetes	3.30	2.18–5.67	<0.001
Obesity or BMI >30 kg/m^2^	4.50	2.22–9.28	<0.001

Hosmer–Lemeshow test *p* = 0.523. BMI, body mass index; CI, confidence interval; OR, odds ratio; SSI, surgical site infection.

**Table 4 jcm-08-00480-t004:** Calculation of independent discrimination for IRIC, ACRI, NNIS risk index, and EuroSCORE-1 (training sample).

Scores	aROC	SE	*p*	95% CI
ACRI	0.67	0.043	<0.001	0.59–0.74
NNIS Risk Index	0.59	0.038	0.024	0.52–0.67
EuroSCORE-1	0.58	0.039	0.045	0.50–0.67
IRIC	0.70	0.039	<0.001	0.63–0.78

ACRI, Australian Clinical Risk Index; aROC, area under the Receiver-Operating Characteristic curve; CI, confidence interval; EuroSCORE, European System for Cardiac Operative Risk Evaluation; IRIC, Infection Risk Index in Cardiac surgery; NNIS, National Nosocomial Infections Surveillance; SE, standard error.

**Table 5 jcm-08-00480-t005:** Calculation of discrimination for the compared scoring systems (test sample).

Scores	aROC	SE	*p*	95% CI
ACRI	0.61	0.057	0.047	0.50–0.72
NNIS Risk Index	0.61	0.052	0.054	0.50–0.71
IRIC	0.66	0.054	0.004	0.55–0.76

ACRI, Australian Clinical Risk Index; aROC, area under the Receiver-Operating Characteristic curve; CI, confidence interval; IRIC, Infection Risk Index in Cardiac surgery; NNIS, National Nosocomial Infections Surveillance; SE, standard error.

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
