# Peer review of "A New Surgical Site Infection Risk Score: Infection Risk Index in Cardiac Surgery"

_jcm, 2019, doi:10.3390/jcm8040480_

Round 1
Reviewer 1 Report
In the manuscript by Bustamante-Munguira et al., titled, "A new surgical site infection risk score: Infection Risk Index in Cardiac surgery," the authors argue for a more predictive model to assess risk of surgical site infection in patients with CABG and/or valve replacement. They found two risk factors that were highly correlative in a retrospective study and had increased ability to identify patients with SSI in a prospective test sample.
To improve the quality of the paper, I have a few minor suggestions:
-Please elaborate on the differences between figure 1 and 2. The last paragraph in the results section does little to address this ambiguity.
-Will the authors please include a small paragraph citing the physiological underpinnings of why they think obesity and diabetes are important risk factors to consider in SSI?
-The strengths and limitations paragraph is too abstract and the wording confuses the reader as to the intent of the writer.
-Page 6 Table 3 - Is the 95% CI for duration of surgery really 1.0-1.0?
-Page 6 Line 192 - The CI for IRIC should be 0.55-0.76.
-Page 7 Figure 1 - The dotted line in the legend does not have a label.
-Page 8 Figure 2 - The dotted line in the legend does not have a label. (aROC)
-Page 9 Line 239 - "more closer" is redundant
Author Response
In the manuscript by Bustamante-Munguira et al., titled, "A new surgical site infection risk score: Infection Risk Index in Cardiac surgery," the authors argue for a more predictive model to assess risk of surgical site infection in patients with CABG and/or valve replacement. They found two risk factors that were highly correlative in a retrospective study and had increased ability to identify patients with SSI in a prospective test sample.
To improve the quality of the paper, I have a few minor suggestions:
-Please elaborate on the differences between figure 1 and 2. The last paragraph in the results section does little to address this ambiguity.
Thank you. The reviewer is right. Following this comment, the last paragraph in subsection Calculation and Construction of IRIC, in Results section, has been rewritten (page 6 lines 194 to 197). The text is now more clear in differentiating Figure 1 that presents independent discrimination for IRIC, NNIS, EuroSCORE-1 and ACRI, from the Figure 2 that presents comparative discrimination (according to Hanley-McNeil test) for IRIC, NNIS and ACRI in the test sample.
-Will the authors please include a small paragraph citing the physiological underpinnings of why they think obesity and diabetes are important risk factors to consider in SSI?
We very thank for this comment. Please, see at the end of the first paragraph in subsection Rationale, Introduction section, the following text has been added (page 2 lines 44 to 47): “Furthermore, special attention should be made of the situation of diabetics and obese patients: 1) diabetics with preoperative hyperglycemia have an increased risk of SSI after cardiac surgery, with an important influence of microcirculatory abnormalities [10], and 2) in obese patients, a decreased blood flow in the adipose tissue is also associated to higher rates of deep SSI after cardiac surgery [7].”
-The strengths and limitations paragraph is too abstract and the wording confuses the reader as to the intent of the writer.
Thank you. The reviewer is right. This paragraph has been rewritten and now is more clear in transmitting goodness and constrains of the new scoring system proposed, in accordance to other available scores.
Please, see the text in subsection Strength and limitations, in Discussion section (page 8 lines 249 to 259):
“Real-world data scoring systems are crucial tools for a most performing management of patients [32]. However, clinicians and researchers need to be aware of goodness and constraints proper of the proposed tools: as a starting point for more rigorous studies, analyses of propension that came from real-world observational sources are destined to improve decision-making process [33], although questions about quality of the evidence may certainly arise [34]. Indeed, observational nature of this study must invite to prudence and temperance, because there would be differences according to the activity in other cardiac surgery centers, even in the same continent. Answering to the current excitation for implementing research into real-world settings, the selection of the most appropriate study design must be accompanied by a clear definition of the intended application for the assessed data [35]: to propose a new SSI scoring system, for which an a priori prospective collected data study was decided [20].”
-Page 6 Table 3 - Is the 95% CI for duration of surgery really 1.0-1.0?
We very thank for this comment. Confidence interval for duration of surgery was 1.00-1.01, but statistical significance was not reached. In the text it was mentioned that non-significant association for duration of surgery was found. Nevertheless, to avoid confusion, the odds ratio and 95% confidence interval for duration of surgery is now added (page 5 line 184). Please, see also Table 3 that only shows variables with significant association: diabetes and obesity.
-Page 6 Line 192 - The CI for IRIC should be 0.55-0.76.
Thank you. The reviewer is right. This was a typo and it has been corrected: compared to NNIS risk index and ACRI, the area under the Receiver-Operating Characteristic curve (aROC) and its corresponding 95% CI for IRIC was 0.67 and 0.55 to 0.76.
-Page 7 Figure 1 - The dotted line in the legend does not have a label.
-Page 8 Figure 2 - The dotted line in the legend does not have a label.
Thank you. In Figures 1 and 2 the dotted line corresponded to reference line, which corresponds to discrimination of 0.5. Please, see the new Figures 1 and 2 where this error is corrected.
-Page 9 Line 239 - "more closer" is redundant
Thank you. The reviewer is right. This error has been corrected.
Reviewer 2 Report
The work by Bustamante-Munguira J, et al titled “A new Surgical Site Infection Risk Score: Infection Risk Index in Cardiac Surgery” was reviewed with great interest. This is an important topic with relevance in cardiac surgery. It’s well know, that post-surgical infections are associated with high mobility, mortality, and increased hospitalization stay, all of these complications have a direct impact on healthcare cost. In the last decade, increased efforts have been made to decrease the rate of these infections, but a recent report from the Centers for Disease Control suggests that some may be increasing despite the use of prophylactic antibiotics and wound vacuums. Therefore, all efforts to identify patients at higher risk to develop these post-surgical infections is key. In this work, authors analyzed several surgical site infection risk factors in both coronary artery bypass graft and valve replacement patients in order to create a new surgical site infection risk score for such individuals. The new score was constructed using a summation system for punctuation using integer numbers, that is, by assigning one point to the presence of either diabetes and obesity. The proposed tool is named Infection Risk Index in Cardiac Surgery (IRIC) and outperforms the other routinely used scoring systems. This work clearly contributes to personalize treatment and care of patients that undergo cardiac or thoracic surgery.
Small edits need to be done to the text (typos and spelling).
Author Response
The work by Bustamante-Munguira J, et al titled “A new Surgical Site Infection Risk Score: Infection Risk Index in Cardiac Surgery” was reviewed with great interest. This is an important topic with relevance in cardiac surgery. It’s well know, that post-surgical infections are associated with high mobility, mortality, and increased hospitalization stay, all of these complications have a direct impact on healthcare cost. In the last decade, increased efforts have been made to decrease the rate of these infections, but a recent report from the Centers for Disease Control suggests that some may be increasing despite the use of prophylactic antibiotics and wound vacuums. Therefore, all efforts to identify patients at higher risk to develop these post-surgical infections is key. In this work, authors analyzed several surgical site infection risk factors in both coronary artery bypass graft and valve replacement patients in order to create a new surgical site infection risk score for such individuals. The new score was constructed using a summation system for punctuation using integer numbers, that is, by assigning one point to the presence of either diabetes and obesity. The proposed tool is named Infection Risk Index in Cardiac Surgery (IRIC) and outperforms the other routinely used scoring systems. This work clearly contributes to personalize treatment and care of patients that undergo cardiac or thoracic surgery.
Small edits need to be done to the text (typos and spelling).
We very thank for this comment. The entire manuscript was revised and edited by Elsevier Language Editing Services. Nevertheless, modifications in this revised version has been carefully revised to avoid typos and spelling mistakes.